# The Development of Traditional Food in Tourist Destinations from the Perspective of Dramaturgy

**Hongwei Mo, Shoubing Yin * and Yunxia Liu**

School of Geography and Tourism, Anhui Normal University, Wuhu 241000, China
* Correspondence: ysblyx@163.com

**Abstract:** The sustainable development of traditional diets in the tourist industry is an important issue. This article analyses the historical evolution and the opportunities for a traditional Chinese dish in tourism. Taking mandarin fish in Anhui cuisine as the research object and the world heritage site Hongcun as the case, this paper investigates the evolution process of the local traditional die driven by tourism and its influence on the construction of gourmet tourism destinations. Primary data were obtained via interview. It was found that, to cater to the mass tourists, the destination created the diet frontstage. Traditional food changed through menu simplification, taste changes, the standardization and scale of production, and the immobilization of presentation. The dieter's frontstage and backstage are not entirely separated; the "middle stage"—Homestay diet is a fusion of them; it is the product of functional differentiation of the frontstage and active integration of the backstage. By expressing the actual state behind the scenes, the middle stage transforms the tourist system from a "supportive experience" to a "peak experience".

**Keywords:** traditional diet; frontstage; middle stage; the smelly mandarin fish





## 1. Introduction

As a component element of tourism experiences, local foods, food experiences, and food tourism have received greater attention in recent years [1]. Local food resources are intrinsically linked to tradition and culture, with a growing interest in culinary heritage, having diverse implications for tourism (re)development and tourist food experience [2,3]. On the one hand, tourism has led to the development of traditional local diets. Unlike other travel activities, eating is essential for visitors to a destination, as it is a primary need [4]). Gastronomy is a significant factor that portrays the culture of destinations [5]; thus, travelers always have a budget for food and beverages, driving the development of destination food industry. On the other hand, tourism damages the authenticity of local diets. Commercialization of tourism can significantly affect local gastronomic identity and image [6], and can result in the deprivation of a "sense of place" for both locals and tourists [7]. Tourist experience demand promotes tourism development, and the literature mainly examines food experiences from the tourist perspective [8], largely neglecting the significance of the local stakeholder perspective on the subject matter, with few exceptions [3,9], and ignoring the opinions of the local subjects. As the producers and providers of food, the actions of local subjects also profoundly impact the construction of gourmet tourism destinations. While tourism development has significantly impacted local diets, there are gaps in the research. Firstly, most culinary tourism and authenticity tourism are Western-centric [10], and most studies focus on the authenticity of the tourist experience. This paper focuses on the perception of the authenticity of Oriental food culture and focuses on the changes in the authenticity of diet under the host's actions.

In China, the drastic social transformation intensifies the modernization and alienation of traditional culture, and the development of many tourist destinations drives the development of local traditional food resources. Some traditional dishes are promoted by

the government as the focus of tourism development and gradually become the symbol of local cuisine. Huangshan City in the south of Anhui Province in China, known as the Huizhou area in ancient times, once produced Huizhou culture with significant influence on Chinese traditional culture. Huizhou culture is known as the epitome of Chinese feudal society due to its integrity and authenticity. Dietary culture is an essential part of Huizhou culture, and now Huizhou cuisine is one of the eight major cuisines in China. Huizhou cuisine has thousands of years of history. The mountainous terrain and lack of food in Huizhou have led to unique dietary varieties and tastes; that is, more pickled food and saltier and spicy tastes. With the spread of Huizhou merchants, Huizhou cuisine was once famous in the Ming and Qing Dynasties; until the beginning of the last century, Huizhou cuisine still had a significant influence in Shanghai, Wuhan, and other places in the Yangtze River Basin. With the decline of Huizhou merchants, Huizhou cuisine also gradually lost its vitality. After the 1940s, it gradually disappeared from the public's view. Huizhou is a mountainous area with sluggish modernization development, and the Chinese once forgot about Huizhou cuisine.

Huangshan began to develop tourism in the 1980s, starting with the development of Huangshan, a world natural and cultural heritage site. In the last 40 years, Huangshan has developed from a mountainous area unknown to outsiders into a famous tourist destination. In 2021, Huangshan City received 63.1686 million tourists, with tourism revenue of 53.81 billion yuan. Many tourists come to Huangshan City, so local food as a tourism support experience has been vigorously developed, and local cuisine as a tourism support experience has thrived. In recent years, the tourism development mode of Huangshan City has gradually changed from sightseeing to experiences, and Huizhou cuisine has gradually become an important tourist attraction that can lead to a peak experience. Huizhou cuisine radiates new vitality, but the new cuisine is different from its original edition. It has formed a few typical dishes represented by the smelly mandarin fish, which sharply contrasts with the traditional Huizhou cuisine system with thousands of dishes.

Huizhou cuisine has contributed a lot to the tourism development of Huangshan City, but tourism has brought significant changes to traditional Hui cuisine. On the one hand, Hui cuisine is once more known to the public. On the other hand, from the perspective of the local population, the new development of Huizhou cuisine has changed its traditional connotation. Locals often say they cannot taste the food they ate when they were children. The purpose of this paper is to make two points. One is to investigate the development and evolution of traditional food in tourist destinations from the host's perspective and how traditional food has become a tourist attraction in the process of local and global confrontation. The second is how the local subject acts to form a beneficial attempt to build a gastronomic tourist destination and the effects of these actions.

## 2. Literature Review

### 2.1. Tourism Food Consumption: Changing the Authenticity of Food Culture

Food heritage is discussed for its potential to raise awareness about the authentic experience of a destination [11]. An increasing number of tourists seek authenticity through tourism dining [10,12,13]. Consumption and enjoyment of local cuisines as a source of embodied experience, meaning, and pleasure [12] have become a way of experiencing other cultures and appreciating the sociocultural characteristics of a destination [11]. A dish is integral to a culture's identity, providing important cultural details [14]. These details are the fundamental indicators of a country's gastronomic identity [15]. Geography and climate affect the products grown in the places where people live, affecting the flavor perceptions of individuals and forming the taste of the society [16], and becoming an essential component of the authenticity of local food culture.

Traditional food has a four-dimensional definition [17]: (1) Production must be mainly local; (2) there must be authenticity (in the recipe, in the origin of the raw materials, or in the production process); (3) the product must have been marketed for an extended period;

and (4) the product must be linked to a gastronomic heritage. When consumers think of traditional foods, they associate them with traditional production methods and recipes, which give them simplicity, natural character, and purity. They value small producers' craftsmanship besides geographical distances in local food [18], since they usually descend from small-scale systems and positively affect the local socioeconomic network [19], and are linked to territory and tradition.

However, unfamiliar foods in the destination may be obstacles to tourists' consumption [20] since eating involves the concrete "incorporation" (ingestion) of things from the environment into the body. Extensive and reliable epidemiological studies have concluded that there is sufficient evidence of human carcinogenicity from eating processed meat [21]. Destinations must adjust food production to meet the needs of tourists to overcome food neophobia as much as possible [20]. For a local cuisine to become a popular attraction in its own right, it has to be filtered through tourism-oriented culinary establishments. Local foods become popular with most tourists only after they are, in some ways and to some degree, transformed [22]. Compared with Western food, which emphasizes the precise ratio of ingredients and seasonings, Chinese food pays more attention to the chef's cooking experience [23], which allows more flexibility for dietary adjustments. These adjustments form the difference between the frontstage and the backstage of the diet, and authenticity becomes a kind of performance for tourists [24]. Avieli's differentiation of diet frontstage and backstage provides a perfect perspective for understanding diet culture authenticity. However, it has not fully applied the process of differentiating the dining place to the development of the food culture in the destination.

*2.2. Dramaturgy: A Perspective to Understand the Authenticity of Food Culture*

Based on the research perspective of the "spatial turn" of sociology and the "cultural turn" of geography [25], researchers began to incorporate space as an essential concept into the research paradigm of tourism reconstruction and change [26]. Goffman's (1959) [27] sociological image of dramaturgical and the metaphor of "life as a theatre" have been used to explore how reality is co-constructed and jointly negotiated through the stage space [28]. MacCannell [29] introduced Goffman's Dramaturgy into tourism research, The staged experience consumed by visitors is considered the frontstage presentation of the culture, and the actual everyday life of the hosts is kept backstage, out of the view of visitors [30]. It was initially created to meet the needs of tourists and to secure the hosts' fragile "real lives" concerning the frontstage and backstage regions [29].

With the development of tourism, the previous perspective of dual separation between frontstage and backstage can no longer explain the current problems. The exhibition at the front stage is not only for the tourists but also for the "backstage" community, which directly impacts the community's cultural attitude and cultural inheritance [31]. Local communities had endured change before tourism commercialization; therefore, cultures may well be able to adapt in certain areas, such as the staging of occasions and events, which reinforces the place and cultural vitality and coherence [32]. Some scholars focus on space when discussing authenticity. Daugstad and Kirchengast (2013) [33] explored how summer farmers in Austria and Norway dealt with their dual roles as farmers and tourist hosts. By adding a pseudo-backstage, the farmers set the terms of access, and at the same time, tourists may perceive that they are receiving special treatment. Theoplacity's union of the objective and subjective, first introduced by Belhassen et al. (2008) [34], focused on the phenomenological proximity of the ideological, spatial, and even temporal authenticity dimensions to the visitor's pilgrimage experience.

The Dramaturgy theory has a new development in sociology. In Goffman's explanation, "frontstage" is a place where various symbolic symbols are used for specific performances, while "backstage" is the content hidden behind the scenes and trying to cover up. The "wall" is the boundary between the frontstage and the backstage, forming two completely separated spaces. The significance is to hide the behaviors that do not meet the expectations in the "backstage", constructing the authority and sanctity of "frontstage",

and creating a sense of distance between the audience and the actors, and strengthening the authority. Joshua Meyrowitz noted the broad application of television, disintegrating the role of the "wall" and exposing "backstage", which also affected people's cognition and behavior. This kind of "backstage" exposure to the "frontstage" is called the "middle stage" [35]. The stars actively expose the "backstage" life scenes, working environment, and emergencies through various shows to form a middle stage [36]. By showing the daily and ordinary nature of the backstage, although it will have a particular impact on the image created by the frontstage, it will also make the audience feel close to it, increasing the recognition between groups [37]. Based on this, the research framework of this paper is shown in Figure 1.

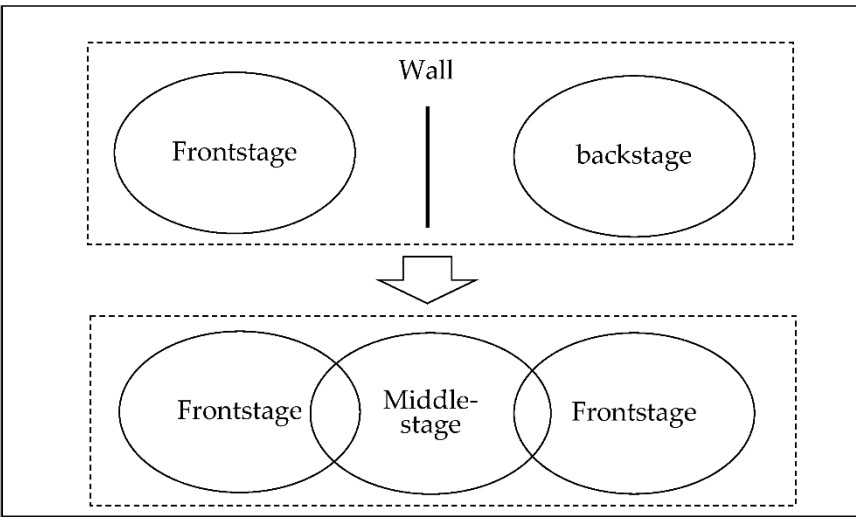

**Figure 1.** The change in the research framework of Dramaturgy.

Avieli pointed out that the heritage site can build tourism recipes by "inventing culinary heritage", introducing foreign dishes, and simplifying local dishes. There is a clear difference between the frontstage tourism recipes and the backstage residents' recipes [24], which shows differences between the frontstage and backstage. In addition, the practice of a tourism diet has a more subjective "initiative", a specific and multi-sensory production practice in constant change; its meaning of implication and expression is richer [23]. Traditional Chinese cooks are more focused on feeling and resistant to quantitative and standardized procedures. Therefore, we need help to grasp the characteristics of cooking. The differentiation of food stage space is an essential clue when studying the influence of tourism on the development of traditional food.

## 3. Case Selection and Data Acquisition

We selected the smelly mandarin fish as the research object, which has formed an industrial chain of raw material breeding, pickling processing, restaurant cooking, cooked food packaging, snack innovation, and even cultural and creative product development. It has become a veritable representative of Huizhou cuisine and even Huizhou culture. On 30 December 2021, Huangshan's catering and cooking industry association released "Big data on the development of the smelly mandarin fish industry, Chinese Huizhou cuisine". These data show that there are 50 processing and pickling enterprises, 6 with an output value of more than CNY 100 million; 8000 people engaged in the pickling work; more than 5000 sellers; 10,000 cooks; and 50,000 employed people. The sales market mainly distributes in the Anhui Province, Beijing, Shanghai, Guangzhou, Hong Kong, and Macao, and abroad mainly distributes in Southeast Asia, Japan, Europe, the United States, and Australia, with an annual output value of about CNY 4 billion [38]. Among the restaurants of Huizhou, the smelly mandarin fish has become the dish the restaurant has to provide.

It is the most influential signature dish and the most popular tourist dish, and thus is a suitable research object to explore the impact of tourism on traditional diets.

Hongcun, an ancient village in southern Anhui, and a world cultural heritage site, is selected as the case site. Hongcun is one of the areas with a complete traditional culture in Huizhou. Since 1997, Hongcun tourism has developed rapidly. In 2018, the number of tourists reached 2.5 million, and the ticket revenue reached CNY 130 million, indirectly driving the tourism industry with a scale of more than CNY 1 billion. We selected Hongcun mainly based on three considerations: (1) The scenic spot retains more locals, and the villagers have not moved out. The core scenic spot has the dual attributes of tourists' play and residents' life. (2) Before its tourism development, Hongcun had the tradition of eating smelly mandarin fish. In the development of tourism, local people also participate in catering services, which ensures the tradition of the local diet. (3) Hongcun is a relatively mature rural tourism destination and a famous gathering place for small tourism enterprises in China. All food suppliers have the smelly mandarin fish as their signature dish.

In the case site Hongcun, we conducted a 24-day field survey from 28 September to 8 October 2020, from 31 August to 6 September 2021, and from 6 to 10 July 2022. We collected data through semi-structured interviews and observation. The interviewed objects include catering operators, community residents, scenic spot managers, village committee staff, and other relevant groups familiar with the development of the smelly mandarin fish. The questions include how the operators produced the original stinky mandarin fish, whether tourists accept the original local taste, whether tourism development has brought changes to the production of this dish, and how local people understand and evaluate these changes. The time of each respondent is about 30 min (Table 1). In addition, researchers experiment with smelly mandarin fish in different types of restaurants and observe the location, style of decoration, style of crockery, and quality of service of different dining venues. The aim was to understand whether the operator adjusted its business strategy with tourism and what actions it has taken between developing the food economy and protecting food culture authenticity.

**Table 1.** Basic information of interviewees.

| Number | Gender | Age | Occupation | Number | Gender | Age | Occupation |
|--------|--------|-----|-----------|--------|--------|-----|-----------|
| T01 | F | 52 | Homestay owner | T18 | M | 51 | Homestay owner |
| T02 | M | 38 | Homestay owner | T19 | M | 30 | Homestay owner |
| T03 | M | 53 | A 20 years Chef | T20 | F | 53 | Restaurant owner |
| T04 | M | 32 | Restaurant staff | T21 | M | 40 | Restaurant owner |
| T05 | F | 43 | Restaurant owner | T22 | M | 47 | Restaurant owner |
| T06 | F | 51 | Restaurant owner | T23 | M | 35 | Homestay owner |
| T07 | M | 48 | Homestay owner | T24 | F | 34 | E-commerce service personnel |
| T08 | F | 36 | Restaurant owner | C01 | M | 67 | Community residents |
| T09 | M | 33 | Homestay owner | C02 | F | 36 | Community residents |
| T10 | M | 53 | Homestay owner | C03 | M | 65 | Community residents |
| T11 | M | 33 | Homestay owner | C04 | M | 70 | Community residents |
| T12 | F | 40 | Homestay owner | C05 | M | 63 | Community residents |
| T13 | M | 42 | Homestay owner | C06 | F | 51 | Community residents |
| T14 | M | 40 | Homestay owner | S01 | M | 55 | Public Security Administrator |
| T15 | F | 39 | Homestay owner | S02 | F | 48 | Marketing Director |
| T16 | M | 40 | Homestay owner | S03 | M | 50 | Village committee staff |
| T17 | M | 45 | Homestay service personnel | S04 | M | 49 | Vice president of the Catering Association |

*Note*: F, female; M, male; T, Tourism service personnel; C, Community residents; S, Scenic spot managers.

## 4. Findings and Discussions

*4.1. State of Diet before the Development of Tourism: Diet Backstage*

The diet backstage is the dining space of residents' homes, which is a space for community residents to make, cook and eat, and it is inaccessible to tourists.

(1) The dietary characteristics of the backstage. Huizhou people live extremely diligently and frugally, and there is a proverb, "cooking slowly and eating slowly", which means "if you want to eat Huizhou food, you must be able to wait". When they cook the fish for themselves, the locals will determine the fire and seasoning by comprehensively considering the relationship between the amount of pickling salt, the pickling time, the control of the fire, and the cooking time. Therefore, rather than buying semi-finished smelly mandarin fish processed by enterprises, locals prefer to marinate it themselves. The quality of marinating depends on personal technique and feeling. Therefore, food behind the scenes shows more diverse tastes, and quality is harder to control.

> *"When we cook the smelly mandarin fish, we first fry it in oil, fry both sides until golden brown, and then simmer, so it takes twenty or thirty minutes to be fully cooked."* (C03). *"Some people may pickle well, while others may not, but it must pickle according to each person's needs."* (T11). *"Although every house has a refrigerator now, we will still eat according to our needs. It will taste a little bit worse after a long time of storage."* (C06)

(2) The shift in the diet backstage. Historically, mandarin fish were not common in residents' daily diets because of its high price. With the development of tourism and the improvement of the economic level, the smelly mandarin fish has gradually become part of the daily diet. Daily consumption has led to the boredom of residents. They continue to choose to eat mandarin fish on important days such as festivals and family and friends' gatherings, forming a return from festivals to daily life and then to "festivals" (including essential days for receiving relatives, friends, and distinguished guests). Tasting smelly mandarin fish also represents a kind of memory and emotional sustenance of past scenes, which is not common in the daily diet.

> *"Although the smelly mandarin fish is so famous now, we seldom ate it when we were young."* (C05). *"We are tired of eating. Now we only eat during the Spring Festival or at a party between friends."* (T14). *"I remember when I was a child, the smelly mandarin fish made by my father was very delicious. At that time, it was not easy to get mandarin fish. It felt good for the whole family to eat together. I haven't eaten any better since."* (T17)

*4.2. The State of Diet Driven by Tourism Development: Diet Backstage*

(1) Simplify the menu and symbolize the traditional diet. The development of tourism has directly led to the commercialization of traditional diets. Compared with the traditional Huizhou cuisine, the amount of diet provided by tourism is much lower. Some traditional specialties with complex production technology, long preparation periods, unstable market demand, and short storage time are gradually declining among them. To meet the consumption expenditure level of tourists, some dishes with higher prices rarely appear on the table in scenic spots. The smelly mandarin fish represent these dishes as they are easy to breed, easy to produce on a large scale, and easy to preserve, gradually occupying the catering market in tourist destinations.

> *"If the guest doesn't come, the prepared ingredients will rot. If the guest comes, you cannot do business without preparation. It's difficult to balance, so I don't provide some dishes = difficult to store."* (T16) *"There are wide varieties and ingredients of traditional Huizhou cuisine, and the cost of some dishes is high, but you can't set the price too high for tourism. Otherwise, tourists will think you bully consumers, so we don't cook many dishes."* (T14)

(2) Adjust the production process to cater directly to customers' tastes. The smelly mandarin fish has been pickled and fermented, and the local flavor is salty and spicy,

making it difficult for tourists to adapt. With the increase in mass tourism, guests show apparent regional differences in taste. Restaurants mainly "improve" the taste by shortening the pickling and fermentation time. When an operator began to adjust the bite and obtain the recognition of tourists, other operators also began to imitate this. Further, this adjustment gradually became conscious behavior.

> *"We used to ask the guests about their tastes because, as a receptionist, you can't always follow your own."* (T03). *"For example, we used to pickle for three days, but now it can be pickled for two days so that the smell of fish will be lighter."* (T07). *"Someone did this, and the guests responded very well. Then we did the same. After all, it's business. How can we keep everything?"* (T09). *"Later, the guests came, as long as we knew where they came from, we would adjust according to the taste of that place."* (T13)

(3) Increasing the production rate affects the richness of taste. Influenced by the increase in tourist demand and the restriction of world heritage sites, restaurants cannot adapt to mass tourists' demand by expanding their restaurants' scale, so they have to "scale" and "standardize" production, which indirectly changes the taste of food. In terms of scale, restaurants have begun to pickle the smelly mandarin fish on a large scale. To avoid waiting too long, restaurants can serve customers more quickly by cooking multiple fish together; however, traditionally, they are cooked one at a time. In terms of standardization, this includes the precise proportion of pickled salt and the compression of cooking time by high-power gas stoves. The traditional smelly mandarin fish must be simmered to make it taste better.

> *"Hundreds of the smelly mandarin fish are all pickled together; otherwise, there will not be enough."* (T22). *"Afterwards, there were too many guests, so we could only cook many fishes in one pot, while it was too time-consuming to cook one at a time.* (T20). *"In the past, we pickled fish by feel, but now hundreds of fishes need to be pickled at a time; if the salt content is inaccurate, pickling failure will cause huge loss."* (T06) *"Now restaurants use high-power gas stoves, the dishes are = not tasty, and we could notice subtle changes when we eat them, the guest certainly couldn't."* (C03)

(4) The form of food on the table and dishes is immobilized. Food is usually prepared in advance; sometimes, it will cool down after being for cooked for a long time. Adding supporting equipment is another change in the smelly mandarin fish that has been implemented to achieve the necessary taste and heating effect. Usually, a small alcohol stove is added under the plate containing fish, and guests can eat while heating. In addition, because the complete mandarin fish is relatively beautiful in form, for the needs of cost control and plate loading form, a mandarin fish dish is generally 400 g. It is loaded in the shape of a complete fish and put in a plate like a fish. The weight and shape of the fish have gradually become the standard, immobilizing the taste of the diet.

> *"It's usually cooked in the morning and reheated when tourists come."* (T04). *"In the past, we didn't use an alcohol stove, but now because we didn't cook it thoroughly, we need to burn it for a while."* (T17). *"A fish in a restaurant is usually 400 g, to control the cost and look good on the plate. In fact, the fish eaten by our local people are sometimes larger, and the taste is better if fish is larger."* (C06)

To meet the needs of tourism, restaurants produce food by expanding the scale and standardizing operations. On the one hand, this promotes the quality assurance of traditional food. On the other hand, the richer expression of the local flavor is limited, forming the phenomenon of taste homogenization that community residents describe as follows: "every restaurant tastes the same". By simplifying the menu, catering to people's tastes, and meeting the demand, the local traditional diet has been accepted by the public and has become a tourism specialty. Therefore, the smelly mandarin fish has become a tourism commodity specially prepared for tourists and has formed a diet frontstage.

We found that the frontstage has the following characteristics. Firstly, it reshapes the image of food that tourists expect. To better explain the traditional characteristics to

tourists, the traditional diet's complex and varied production process is abstracted into a standard production process. Secondly, it caters to tourists' tastes exclusively, and the food taste is completely based on the tourists' ideas instead of highlighting local differences. Thirdly, to receive more guests, the production scale is expanded and the taste standardized, which destroys the traditional diversity of flavor. The food at the frontstage has become the product of tourist demand, with apparent homogeneity.

### 4.3. Integration of Frontstage and Backstage Has Been Driven by Transformation and Upgrading of Tourism Development: "Middle Stage"

#### 4.3.1. What Is the Diet "Middle Stage"?

(1) Concept definition. The food middle stage is the diet space formed by the party operator's dynamic adjustment of the food production mode in frontstage and integration of authentic food elements from backstage. In Hongcun, this kind of space is highlighted in some Homestays. The middle stage is a negotiation space between the over-commercialization of the frontstage and the authenticity of the frontstage.

(2) Boundary distinction. There is a boundary between the "middle stage", the "frontstage", and the "backstage". First of all, the diet middle stage does not receive mass tourists. We found a lot of Homestays in Hongcun with signs at the door containing the following information: declined to visit, not open to the public. Additionally, in some Homestays, meals are only provided to the guests who are staying overnight; only visitors using the accommodation can eat there. This is different from the frontstage, which is open to all visitors, and it is also different from the backstage, which is difficult for visitors to reach.

(3) Quantity. The number of the middle stage is minor. In the Hongcun field research, we conducted a sample survey. There are more than 400 diet and accommodation business operators in Hongcun. According to the proportion of "1/10", we sampled 40 business establishments and found that only four met the conditions of "middle stage ", and only the Homestay space met the characteristics of the middle stage.

(4) Formation time. Since Hongcun became a world heritage site in 2000, tourism has developed rapidly. In 2000, Hongcun had 80,000 visitors; in 2006, 700,000; in 2013, 1.5 million; and in 2017, it exceeded 2 million visitors and continued to 2.76 million in 2019. (Since 2020, due to the impact of COVID-19, the number of tourists dropped below 2 million). With the rapid increase in tourists, Hongcun's diet and home accommodation industries have developed rapidly. After opening the Huangshan high-speed railway in 2015, Hongcun has become the peak period for tourists and foreign capital to enter. From 2015 to 2018, the development of Homestays in Hongcun entered the peak period, with many Homestays operating at this time. The former Hongcun Homestay business scope includes diet reception in the courtyard and accommodation reception in the room, similar to small hotels' development. Starting in 2017, some Homestay began to translate into the middle-stage place for niche tourists.

#### 4.3.2. How Is the "Middle Stage" Formed

Diet "middle stage"—the frontstage differentiation and the backstage's active integration. With the increase in tourists engaging in leisure vacations and cultural experiences, the stage of separating the diet of the mass and niche tourists occurs, and there is a local integration between the frontstage and the backstage, forming a "middle stage" dietary space in Homestay. First, considering the niche tourism market or the limitation of their reception capacity, some Homestays began to change their business model, reposition their target consumers, and gradually withdraw from the diet market for non-lodging guests. Second, to improve the service quality, reducing the mass tourist market provides more preparation time for catering and strengthens the communication between hosts and guests. Third, the "standardized" production of the frontstage awakens the local people's differentiated consciousness of the "tradition" of diet.

*"When guests come, we recommend them to the restaurant. We can't do both homestay and catering (non-lodging guests). Cooking fumes also have an impact on guests."* (T18).
*"In fact, there is no authentic smelly mandarin fish. Those restaurants are well publicized online. What we eat at home is authentic the smelly mandarin fish."* (T14)

(1) The expression of the authentic taste of the backstage diet. The first is non-standardized production. The scale and frequency of Homestay diet production are lower than those of restaurants. The producer is not a professional chef, and the characteristics of food heterogeneity are apparent. For the needs of tourism services, Homestay diet producers consciously improve their food production skills and make their food quality higher than that of community residents to ensure the quality standard and show differentiation simultaneously. The second is the slow-paced cooking, with few guests and sufficient time. The processing of only cooking one fish at a time fully reflects the tradition of Huizhou cuisine pays attention to heat, and comprehensively presents the richness of dietary taste. Moreover, it is to convey the natural, local taste. The Homestay producers will not blindly cater to tourists but try to convey the salty and spicy real taste of Huizhou cuisine as much as possible. Finally, the expression of food culture, through actively telling tourists about local traditional tastes and food culture, promotes tourists' acceptance of local flavors.

*"The chefs in those restaurants have to cook hundreds of dishes daily. It's like mechanical operation. It requires emotional investment to cook the dishes. "For example, the taste will be better if we are in a good mood and hum a tune and cook the dishes."* (T06) *"Tourists living in our house, as long as they eat here, we will simmer fish for them."* (T08) *"Some tourists don't adapt to our local tastes. We will communicate with them to show the characteristics of the diet and explain well, so some of them can accept."* (F13)

(2) The presentation of dietary practice space from backstage. The first is the presentation of the kitchen. In the restaurant at the front stage, the kitchen is a ban for public tourists, and the Homestay can give tourists access to the kitchen. The host teaches the tourists the skills of pickling mandarin fish and cooking, and they can also provide a kitchen for tourists to cook in, which increases the tourist experience. The second is the presentation of the "home" atmosphere. The dining environment of the Homestay is more family oriented. Tourists can communicate with the host when eating, regardless of the restriction of eating time. In their spare time, the host will even eat with the guests. To some extent, this is a return to the traditional Huizhou family entertaining distinguished guests with the smelly mandarin fish.

*"Tourists sometimes participate in cooking. It's no different from being at home."* (T04).
*"Some tourists have more strange food tastes. The restaurant can't provide it, only we can provide"* (T15). *"Some repeat customers come every year. Sometimes they eat and drink together and talk about family affairs."* (T09)

### 4.3.3. The Influence of the Diet Middle Stage

The expression of the smelly mandarin fish's authentic taste backstage and the practice space's presentation provides tourists with the opportunity to deeply experience cultural authenticity, which has impacted the "most authentic" and "most distinctive" food image created by the frontstage. However, it also cultivates a deeper passionate connection between the host and the guest in relation to food, forming a closer emotional community and allowing niche tourists to achieve the peak experience of tourism food. The dietary practice of tourism provides a realistic environment for people to contact each other. Therefore, the diet, culture, and taste of tourists' participation in the performance harm the host's food habits.

*"The generation like my children is less able to eat salty because we also eat fish cured for tourists, so the taste is light."* (T10)

The emergence of the middle stage is not only the promotion of tourism demand, but also the promotion of local cultural confidence. The locals started to rethink the dietary

practice on the front stage. They began to consider the authenticity of the diet, which is not just for the front's signature image. It should be backstage, in the countryside, and in residents' daily diet practice. Then, locals began to express the local heterogeneity and diversity of dietary elements. However, it seems that once the production is expanded, the food will no longer have the ultimate local flavor. They believe they need to preserve the traditional flavor of food by limiting the scale of production.

### 4.4. Discussion

We find that in tourism development, the frontstage and backstage are not entirely separated, but include the space of the "middle stage", which is the transition zone between the front and backstage. It is the product of the transformation from sightseeing to vacation tourism. However, the middle stage is not a universal phenomenon but a particular substitute for meeting the cultural experience of tourists because tourists want to experience the factual background, so local operators develop the space. The negotiation space between the host and the client is in the middle stage due to the significant information gap frontstage. With the rapid development of tourism in China, more and more homogenization phenomena have emerged, and not only in the diet. The reason is that the tourist destination excessively caters to the needs of tourists and ignores the local discourse and characteristics. The formation of the middle stage resulted from the change in tourists' demands and local people's consciousness. The middle stage provides a better opportunity for interaction between the frontstage and backstage, between the visitors and the host. However, the number of middle stages is small, reflecting that the communication and interaction between hosts and guests of the destination are still inadequate. For example, according to a survey on Hongcun, a world cultural heritage site, many tourists spend less than one day visiting it. However, in the minds of local people and those familiar with Hongcun, it is a treasure trove of culture that takes a great deal of time to understand; this has led to many tourists saying there is nothing to see when visiting Hongcun and feeling disappointed with the village. The same is true of the diet. Many tourists say that the smelly mandarin fish in Hongcun is not unique after eating in the restaurant at the frontstage, and that it is not even as delicious as the smelly mandarin fish in tourists' hometowns. These visitors, who are not exposed to the wonders of backstage eating, assume that the diet in the country of origin is nothing more than that. Therefore, it is necessary to have such a Homestay space as the middle stage to communicate with hosts and guests. If these people live in Hongcun and find such a middle-stage space, they can also experience different destinations' food culture. Of course, Homestay space is only a form of the central district. In our opinion, the middle stage, in the real sense, is to change the homogenization result caused by the excessive performance of the frontstage, and also to modify the performance capability lost due to the excessive background protection. It negotiates between host and guest, which is favorable for the sustainable development of the relationship between host, guest, and destination.

### 5. Conclusions

There are three stages of development and evolution in the local traditional diet driven by tourism, including the stage of pioneer tourists' diet, the stage of mass tourists' diet, and the stage of diet separation of mass and niche tourists. The diet production in the frontstage and backstage is differentiated. The frontstage is a dietary space specially provided for tourists, and the restaurant is the primary carrier. The local traditional diet has gradually transformed into a tourist signature dish through the construction of expected images, catering to the tastes of tourists and production with maximum benefits. The diet backstage is the home of residents, which experienced the "festival"—daily—"festival" evolution process.

The diet frontstage and backstage are partially separated. The frontstage has an impact on the backstage in terms of shaping the image of local food and taste standardization, and the backstage has an impact on the frontstage in conveying a more diversified dietary

culture and taste to tourists. The study found that the Homestay is a kind of "middle stage" of diet, which is the active integration of the backstage and the frontstage. Through the expression of the real taste backstage and the presentation of the diet practice space, the Homestay actively "exposed" the actual state of the local diet, impacted the so-called "most authentic" and "most characteristic" authoritative images frontstage, and also shaped a deeper emotional connection with the host and guest about diet, contributing to the transformation of tourism diet from "supportive experience" to "peak experience".

The differentiation between the diet frontstage and backstage is the production of mass tourism development, and the integration of the frontstage and backstage are the results of the increase in the demand for niche tourism and the local people's reflection and adjustment in dietary practice. In tourist destinations, the "middle stage" is a carrier of the integration of the frontstage and backstage. It is an essential means of sustainable development of traditional diets in tourist destinations and plays a vital role in constructing gourmet tourism destinations. It requires awakening the local subject's consciousness and destination construction to pay more attention to the host's opinions.

**Author Contributions:** Conceptualization, H.M. and Y.L.; methodology, S.Y.; validation, S.Y. and Y.L.; formal analysis, H.M.; investigation, H.M.; data curation, H.M.; writing—original draft preparation, H.M.; writing—review and editing, H.M., Y.L. and S.Y.; visualization, H.M.; supervision, S.Y.; project administration, S.Y.; funding acquisition, S.Y. All authors have read and agreed to the published version of the manuscript.

**Funding:** This research was funded by Science Foundation of Ministry of Education of China (No. 22YJAZH136).

**Institutional Review Board Statement:** Not applicable.

**Informed Consent Statement:** Not applicable.

**Data Availability Statement:** The primary data is obtained through interview; Publicly available data were used in this paper (listed in the References).

**Conflicts of Interest:** The authors declare no conflict of interest.

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
