# Peer review of "The Development of Traditional Food in Tourist Destinations from the Perspective of Dramaturgy"

_sustainability, doi:10.3390/su142416900_

Round 1
Reviewer 1 Report
Thank you for sharing this interesting study.
Proof well for grammar and word choices e.g., "This article analyses the historical evolution and the opportunities that a traditional Chinese dish which from a traditional one to a signature driven by tourism, and its new tendency in the world heritage site." and to be clear and specific e.g., "Main data were obtained by interviewing" - for example, Qualitative data was collected via interviews with .... AND " it has brought some damage to the authenticity of local diet" - aim to be clear and specific so the meaning of the sentences/content is not reduced or creates confusion. AND e.g. " Obviously, the development of tourism pays more attention to the experience and 30 perception of tourists," - is it obvious? AND"of stuff" - poor word choice
Try to use current sources "A dish is an important part of a culture's identity, providing important cultural details (Montanari, 2006)" AND (Jordana , 2000)
(ANHUINEWS, 2021) - All caps
"After 1997, Hongcun tourism has developed rapidly. In 2018," - what is happening now? 2.5 million visitors may not be a lot for Chinese sites. In this and other parts of the paper, be sure to offer data that supports the point being made.
This is vague ". Through 204 reading relevant literature and collecting relevant materials, establish the research plan, formulate and optimize the interview outline." - vague. Offer specifics so the content adds value to the paper and informs readers. What did the observation protocol include - what was the observation process? " The observation contents include dining places, catering dishes, dining environment, supporting facilities, etc." These are locations so be specific rather than adding etc.
It is suggested the rewording and restructuring of this long sentence would aid reader comprehension " In addition, personally consume and experience the smelly mandarin fish, and observe the environmental atmosphere, decoration style and tableware style of different dining places; Went to the local mandarin fish supply market, Huangshan Chinese Huizhou cuisine Museum and other places, and collected the literature on food culture, commodity information and consumption evaluation related to the smelly mandarin fish on the main online review websites as auxiliary data."
The results should aim to present only results - consider if this is needed in the results section "According to Goffman, the backstage is related to the “performance” of the frontstage, but inconsistent with the impression promoted by it. Specifically, the backstage is the dining space of residents' home, which is a space for community residents to make, cook and eat, and is inaccessible to tourists."
There are some interesting results. the way the section is written dies not highlight this well. Consider rewording and structuring to strengthen the findings.
How were responses analysed? Thematic analysis? What are the key themes. Greater detail to this may increase the contribution of this paper.
This is an interesting study. Take time to fully develop the paper to reflect this. Attention to details, specificity and proof reading may increase the value of the paper.
Author Response
Thank you very much for your pertinent comments. I have revised the whole text.
Please see the attachment,I have marked the modified part in yellow.
look forward to your further comments.

Reviewer 2 Report
The topic of an innovative approach to building a modern tourist product in Huangshan City, based on the traditional Chinese dish, is very interesting, but the development of traditional food in tourist destinations should start from the analysis of the final recipient of tourism production: people (tourists ore guests).
It would be suitable an introductory section (or a new paragraph) on the following topics:
- the shift from the overall tourism product to the tourism experience; - the role of traditional food in tourist destination image building, stressing the shift from a destination product to a destination experience, and from destination marketing to destination management; - general situation of Huangshan City tourism;
These aspects should be highlighted in the introduction and conclusions.
Author Response

(The authors gave the same response as above.)

Reviewer 3 Report
Τhe article is about an interesting topic, namely how emblematic dishes of local cuisines are affected by their fame as touristic attractions. The paper adopts the terminology "backstage" for the behavior of the local society at home, "frontstage" for the behavior of the local society in the touristic market and "middlestage" for the behavior of local society in a form of tourism that requires some explanation. While frontstage and backstage can be well-understood as economic and social models and are well separated from each other, the middlestage situation needs to be explained for the reader's convenience.
The paper relies on a set of interviews of local people. I am not sure that the extracts of the interviews that are presented in the paper correspond to all the conclusions. For instance, the "backstage" interviews say that today local people eat the smelly mandarin fish (smf) only on particular occasions and that they cook it in the traditional way. How is that different from the original situation, i.e., before tourism? Probably today they can afford smf easier but otherwise, these locals seem to consume it as they traditionally did. The paper however says "Therefore, we can see that the diet backstage has also been affected by the develop- 332 ment of tourism and daily modernization. The diet customs and culture have changed, 333 but the frontstage has always fixed the image of diet in the state before the development 334 of tourism." (page 7).
The middlestage situation is praised but it seems to affect local cuisine in an irreversible way. While the frontstage/backstage situation did not eventually affect recipes and dietary habits, this middlestage situation transforms the locals because it penetrates their homelives.
In fact, I would expect more from this paper. I would expect some numerical/statistical data about how many people are engaged in tourism, since when, how local production was affected because of tourism, where do ingredients come from, when exactly this "middlestage"model appeared, how many homes have switched to the "middlestage" model...The paper relies on interviews with locals and little of solid statistical information is provided.
Author Response

(The authors gave the same response as above.)

Round 2
Reviewer 1 Report
Thank you for your work in editing this manuscript and re-sharing for review. There are interesting findings and useful insights.
Clarity is important and each sentence needs to be clear and add value. for example, "Sustainable development of traditional diet under tourism is an important issue" - first sentence in the abstract is unclear. It could be reworded to make the point clear or deleted.
Proof reading is critical e.g., "This article analyses the historical evolution and the opportunities that a traditional Chinese dish [ this next part does not make sense .... which from a traditional one to a signature driven by tourism, and its new tendency in the world heritage site .... ]". This sentence could be improved with rewording and dividing into 2 sentences as two topics are introduced.
And e.g., "Local food resources are inherently related to tradition and culture, with a growing interest in culinary heritage [remove comma], having diverse implications for tourism (re)development and tourist food experience.."
And "On the one hand, The development of tourism has led to the development of local traditional diet [is this correct? Has tsm led to a local diet? Changed local diets?] , on the other hand, it [what is it?] damages the authenticity of local diet(Athena et al., 2012)[new topic] and globalisation and localisation are two major forces influencing the development of destination food." - there is a lot being aid in this sentence. When sentences are long and complex they become confusing.
Two sentences? "Although tourism development has had a significant impact on local diets, there are gaps in research [full stop?],this study aims to address two research gaps." or consider rewording for clarity.
The aim of the study, findings and lit review start to develop a satisfactory argument. The readability and comprehension of the content is reduced due to the writing quality. Take the time to prepare the paper in a highly professional manner that showcases the interesting and useful research undertaken.
Author Response
Thank you very much for your valuable comments on the modification. It has been edited and modified. Looking forward to your further reply.

Reviewer 3 Report
The text has been improved in terms of clarity of information and logical structure but it should be edited intensively.
Author Response

(The authors gave the same response as above.)
